# Identification of MicroRNAs Associated with Prediabetic Status in Obese Women

**DOI:** 10.3390/ijms242115673

**Published:** 2023-10-27

**Authors:** Leona Kovac, Thilo Speckmann, Markus Jähnert, Pascal Gottmann, Louise Fritsche, Hans-Ulrich Häring, Andreas L. Birkenfeld, Andreas Fritsche, Annette Schürmann, Meriem Ouni

**Affiliations:** 1Department of Experimental Diabetology, German Institute of Human Nutrition Potsdam-Rehbruecke (DIfE), 14558 Nuthetal, Germany; leona.kovac@dife.de (L.K.); markus.jaehnert@dife.de (M.J.); pascal.gottmann@dife.de (P.G.); meriem.ouni@dife.de (M.O.); 2German Center for Diabetes Research (DZD), 85764 Neuherberg, Germany; louise.fritsche@med.uni-tuebingen.de (L.F.); hans-ulrich.haering@uni-tuebingen.de (H.-U.H.); andreas.birkenfeld@med.uni-tuebingen.de (A.L.B.); andreas.fritsche@med.uni-tuebingen.de (A.F.); 3Institute of Diabetes Research and Metabolic Diseases (IDM) of the Helmholtz Center Munich, University of Tübingen, 72074 Tübingen, Germany; 4Department of Internal Medicine IV, University Hospital Tübingen, 72076 Tübingen, Germany; 5Institute of Nutritional Sciences, University of Potsdam, 14558 Nuthetal, Germany

**Keywords:** miRNA, prediabetes, obesity, insulin signaling

## Abstract

MicroRNAs (miRNAs) recently emerged as means of communication between insulin-sensitive tissues to mediate diabetes development and progression, and as such they present a valuable proxy for epigenetic alterations associated with type 2 diabetes. In order to identify miRNA markers for the precursor of diabetes called prediabetes, we applied a translational approach encompassing analysis of human plasma samples, mouse tissues and an in vitro validation system. MiR-652-3p, miR-877-5p, miR-93-5p, miR-130a-3p, miR-152-3p and let-7i-5p were increased in plasma of women with impaired fasting glucose levels (IFG) compared to those with normal fasting glucose and normal glucose tolerance (NGT). Among these, let-7i-5p and miR-93-5p correlated with fasting blood glucose levels. Human data were then compared to miRNome data obtained from islets of Langerhans and adipose tissue of 10-week-old female New Zealand Obese mice, which differ in their degree of hyperglycemia and liver fat content. Similar to human plasma, let-7i-5p was increased in adipose tissue and islets of Langerhans of diabetes-prone mice. As predicted by the in silico analysis, overexpression of let-7i-5p in the rat β-cell line INS-1 832/12 resulted in downregulation of insulin signaling pathway components (*Insr*, *Rictor*, *Prkcb*, *Clock*, *Sos1* and *Kcnma1*). Taken together, our integrated approach highlighted let-7i-5p as a potential regulator of whole-body insulin sensitivity and a novel marker of prediabetes in women.

## 1. Introduction

Since the prevalence of type 2 diabetes (T2D) continues to rise, efforts to determine individual disease risk are increased, as early interventions could reduce or prevent secondary complications. Obesity and insulin resistance are potent drivers of T2D onset, underpinned by genetic predisposition [1]. Studies looking at familial T2D have demonstrated an average estimated heritability of 30–70% when considering all age groups [2,3,4,5], with lower rates after the age of 60 years [2]. While a significant number of genetic loci associated with T2D risk were identified through large-scale meta-analysis of genome-wide association studies (GWAS), only a fraction of single-nucleotide polymorphism (SNP) based T2D heritability could be explained by these genetic variants [6]. Beyond genetics, epigenetic changes may be one factor in the “missing heritability” of T2D. Epigenetics describe modifications that affect gene expression without alterations in genomic sequence, including DNA methylation, histone modifications and non-coding RNAs [7]. MicroRNAs (miRNAs) are a class of non-coding RNAs that serve as regulators of gene expression [8]. Most miRNAs induce mRNA degradation and translational repression through interaction with 3′ untranslated region (3′ UTR). However, some miRNAs can interact with 5′ UTR, coding and promoter regions, and under certain conditions can induce translation [8]. miRNAs can be found extracellularly in blood plasma where they remain stable because they are protected from degradation by RNA-binding proteins (e.g., Argonaute) [9] or encapsulated in extracellular vesicles (EVs) [10]. In recent years, it has been shown that circulating miRNAs can serve as means of communication among different insulin-sensitive tissues, mediating diabetes development and progression [10,11]. As such, miRNAs are a valuable proxy for epigenetic alterations associated with T2D.

Here, we hypothesized that circulating miRNA can be used as markers of impaired fasting glucose, which is part of the definition of prediabetes. Therefore, we sought to identify differences in miRNA levels in plasma between women with elevated fasting glucose concentrations (IFG) and those with normal glucose tolerance (NGT). We applied a translational approach by integrating human and mouse data from female New Zealand Obese (NZO) mice whose diabetes susceptibility can be predicted based on blood glucose and liver fat content [12] (Figure 1a). Several miRNAs were significantly altered in plasma of women with prediabetes and in metabolically relevant tissues (pancreatic islets and white adipose tissue) of diabetes-prone NZO mice. Finally, we demonstrated the ability of one of the candidate miRNAs (let-7i-5p) to regulate the expression of insulin signaling pathway components in β-cells.

## 2. Results

### 2.1. Identification of Six miRNAs in Plasma of Women with Prediabetes

In order to identify circulating miRNA markers of prediabetes we used plasma samples from a subgroup of women participating in the TÜF (TÜbingen Family) study with either normal glucose tolerance (NGT, N = 44) or impaired fasting glucose (IFG, N = 20) (Figure 1a). Age, BMI and blood glucose 120 min after glucose bolus were not different between NGT and IFG, while fasting blood glucose was significantly higher in the IFG group (*p* < 0.0001) (Table 1). Out of 384 tested miRNAs, 172 were detected in plasma of women with NGT and/or IFG, and six were significantly different between NGT and IFG (miR-652-3p, miR-877-5p, miR-93-5p, miR-130a-3p, miR-152-3p and let-7i-5p). All miRNAs were present at a higher level (Figure 1b) in the IFG group. Let-7i-5p and miR-93-5p were significantly correlated with fasting blood glucose levels (Figure 1c), and no significant correlations were found for age, BMI and blood glucose 120 min after glucose bolus. Once tested by a generalized linear model, neither BMI nor age showed a significant association to plasma levels of miR-652-3p, miR-877-5p, miR-93-5p, miR-152-3p, miR-130a-3p or let-7i-5p (Appendix A).

### 2.2. Validation of Candidate miRNA Expression in Metabolically Relevant Tissues

In order to assess whether levels of detected miRNAs in human plasma are altered in metabolically relevant tissues in the prediabetic stage, we used data of female New Zealand Obese mice; an obese model which develops insulin resistance and diabetes depending on early blood glucose concentration and hepatic lipid storage [12,13]. MiRNA-sequencing data from pancreatic islets and white adipose tissue (WAT) from 10-week old diabetes-resistant (DR) and diabetes-prone (DP) mice that did not reach hyperglycemia at this age (Appendix A) were compared to plasma profiles of NGT and IFG women. DP mice exhibited slightly higher body weight and liver fat content than DR mice (Appendix A). Of the six miRNAs identified as increased in women with prediabetes, four were expressed in mouse islets and three in WAT. The four islet miRNAs (miR-93-5p, miR-130a-3p, miR-152-3p and let-7i-5p) were highly elevated in the DP in comparison to the DR group. In WAT, two miRNAs were elevated (let-7i-5p, miR-652-3p), and miR-130a-3p was lower abundant in DP mice. Only let-7i-5p, was found in both islets and WAT and reflected the findings in plasma of women with IFG (Figure 2). Additionally, let-7i-5p is the only miRNA which is 100% conserved between human and mouse. Even though miR-130a-3p, miR-152-3p and let-7i-5p exhibit high level of conservation (96%, 97% and 91%, respectively), we and others previously demonstrated that a single nucleotide polymorphism can alter the binding to the target mRNA [14]. Hence, we hypothesized let-7i-5p is the most promising potential prediabetes marker.

### 2.3. Increased Expression of let-7i Reduces Expression of Genes Involved in the Insulin Signaling Pathway

Aside from potential markers, we wanted to assess whether the prediabetes-associated let-7i-5p miRNA has biological functions associated with T2D progression. We first identified 6214 differentially expressed transcripts between DR and DP female NZO mice via whole islet RNA-seq [12], and compared these to predicted let-7i-5p targets. There was an overlap of 2333 genes downregulated in DP mice [12], of which 437 were experimentally validated in humans, according to the databases [15]. These 437 candidates were further selected for KEGG pathway analysis (Figure 3a). There was a strong enrichment for, among others, pathways relating to GABAergic synapse, circadian rhythm, PI3K-Akt and insulin signaling. Of note, genes involved in insulin signaling were overrepresented in several other identified pathways. For example, *Prkcb* was found not only in insulin signaling, but also in the GABAergic synapse, mTOR, thyroid hormone signaling and choline metabolism. Therefore, we focused on insulin signaling genes for the in vitro validation.

To investigate the role of let-7i-5p in islet cells, we induced its overexpression in the rat β-cell line INS-1 832/12 (Figure 3b). Let-7i-5p overexpression resulted in decreased expression of several predicted let-7i-5p target mRNAs implicated in the insulin signaling pathway, with conserved seed sequences in human, mouse and rat (Appendix A). In contrast, the theoretical targets Foxo1 and Itpr3 were unchanged upon let-7i-5p overexpression. Taken together, the in vitro experiments suggested that the elevated expression of let-7i-5p might be associated with the impairment of insulin signaling in pancreatic β-cells (Figure 3c).

We speculate that differentially expressed transcription factors (TFs) between DP and DR are putative regulators of let-7i-5p in islets. Therefore, we screened for TFs with altered expression and predicted binding sites within the let-7i-5p promoter. Four TF-encoding genes fulfil our in silico criteria and correlate with let-7i-5p levels; *E2f2*, *Patz1*, *Plagl2* and *Sp4* (Appendix A).

## 3. Discussion

In the current study, we provide a panel of miRNAs altered in plasma of women with impaired fasting glucose, integrated with tissue-specific alterations of miRNA levels in diabetes-prone NZO mice. To the best of our knowledge, only one study, performed in a South African population, investigated changes in circulating miRNA levels associated with prediabetes specifically in women [16]. The majority of available studies are performed on males or mixed-sex cohorts [17,18,19,20,21,22]. However, a recent study highlighted sex specificity of the visceral adipose tissue miRNA signature associated with prediabetes and diabetes [23]. Therefore, the knowledge about female-specific markers of prediabetes is lacking. Another common limitation of studies dealing with circulating miRNAs is the absence of connection to relevant tissues. Here, in addition to the identification of markers associated with elevated fasting blood glucose in women, we recognized a potential source tissue and suggested a mechanism through which let-7i-5p can modulate insulin sensitivity in pancreatic islets.

We first identified six miRNAs which were significantly increased in plasma of women with prediabetes, compared to women with normal glucose tolerance. Some of these miRNAs have previously been linked to insulin resistance. For example, miR-93-5p has been suggested to negatively regulate GLUT4 expression in kidney and fat cell lines [24], and miR-152-3p plasma levels were associated with development of diabetic nephropathy in individuals with T2D [25]. Besides miR-152-3p, elevated expression of miR-130a-3p was also detected in islets of hyperglycemic donors. Via targeting of pyruvate dehydrogenase E1 alpha (PDHA1) and glucokinase (GCK), their overexpression in INS-1 cells reduced intracellular ATP levels in beta cells [26]. In a screening of miRNAs in abdominal subcutaneous WAT of obese insulin-resistant, obese insulin-sensitive and lean women, miR-652-3p was identified and, via targeting of genes related to insulin signaling, its overexpression increased insulin-stimulated lipogenesis [27]. The amount of miR-652-3p in extracellular vesicles was transiently elevated after acute exercise, and this effect was associated with insulin sensitivity [28]. MiR-877-5p has not been described in relation to glucose homeostasis or insulin sensitivity. It is a marker for different types of cancer [29,30] and was identified as potential synaptosomal miRNA upregulated with progression of Alzheimer’s disease [31]. Of note, miR-23a [22], miR-148b-3p and miR-27a-3p [32], which have previously been implicated in prediabetes, were not differentially abundant in our study, possibly due to sex differences since the cohorts used in the other studies were largely male. Regarding a possible influence of sex hormones in our female cohort, we found indications that estrogen signaling may impinge on the expression of the candidate miRNAs. For example, miR-93-5p expression was decreased in cases of estrogen deficiency [33,34], whereas miR-152-3p was associated with lower estrogen in female offspring [35]. Conversely, the candidate miRNAs may be implicated in regulating estrogen receptor signaling themselves. Indeed, the expression of several miRNAs were inversely correlated with levels of estrogen receptor in human breast cancer (miR-93-5p [36,37]; miR-130a-3p [38,39,40]; and let-7i-5p [41]), and estrogen receptor α was shown to be a direct target of miR-130a-3p [42,43]. Furthermore, estrogen production or secretion was decreased by let-7i [44] or miR-130a-3p [33], respectively.

In human studies, it is difficult to link altered patterns of circulating miRNA to the responsible tissue. Therefore, we designed a translational approach where we chose a mouse model which resembles the human metabolic syndrome. The NZO mouse is a model of polygenic obesity and shows a T2D-like phenotype. In contrast to male mice, diabetes susceptibility of NZO female mice is heterogenous. When fed with 60% high fat diet, DP mice showed a rapid increase in blood glucose starting from the age of 8 weeks and become diabetic at 16 weeks, whereas DR exhibited moderate increase in blood glucose concentration and remained normoglycemic until the age of 25 weeks. We previously established prediction criteria, based on the hepatic fat content and blood glucose levels, to distinguish between DP and DR before the onset of T2D [12]. Female 10-week-old NZO mice can therefore be used as a model for prediabetes suitable for our translational approach.

In order to estimate from which tissue the six miRNAs are released into bloodstream, we assessed their levels in pancreatic islets and WAT of DR and DP mice. We found miR-93-5p, miR-152-3p, miR-130a3p and let-7i-5p to be significantly higher abundant in islets, while let-7i-5p and miR-652-3p were higher and miR-130a-3p lower in WAT of diabetes-prone mice. Presumably, let-7i-5p is the most promising marker candidate because it correlated with fasting blood glucose levels in women and both mouse islets and WAT exhibited a similar pattern of expression. So far, no data described the role of let7i in pancreatic islets in the progression of T2D. However, a previous study showed that let-7 family regulated the expression of components of the INSR/IGF1 pathway in individuals with pancreatic ductal adenocarcinoma [45]. Finally, in vitro experiments in which let-7i-5p was overexpressed in the INS-1 rat β-cell line indicated that this miRNA is a negative regulator of several genes involved in the insulin signaling and secretion pathway, including *Insr*, *Rictor*, *Prkcb*, *Sos1*, *Kcnma1* and *Clock*. A downregulation of *Clock* and *Insr* was additionally observed in the WAT of DP mice (unpublished data). This finding is in accordance with previous studies showing that plasma let-7i-5p is decreased in response to bariatric surgery [46] and exercise [47]. In humans, let-7i-5p was found to be expressed in the majority of human tissues, including adipocytes, liver, skeletal muscle and pancreas [48], supporting the hypothesis of its involvement in the regulation of whole-body insulin sensitivity.

MicroRNAs have been suggested as means of communication among insulin-sensitive tissues participating in maintenance of glucose homeostasis [11,49]. Therefore, we postulate that islet let-7i-5p overexpression contributes to systemic impairment of insulin sensitivity via negative regulation of insulin signaling, leading to defects in glucose homeostasis.

## 4. Materials and Methods

### 4.1. Human Participants and Ethics Statement

A subgroup from the German cross-sectional TÜbingen Family study for type 2 diabetes (TÜF) for people at high risk of developing T2D was used. TÜF currently comprises more than 3000 non-related individuals at increased risk for type-2 diabetes, i.e., subjects with a family history of type 2 diabetes, a BMI > 27 kg/m^2^, impaired fasting glycaemia and/or previous gestational diabetes [50]. Participants’ characteristics are summarized in Table 1. In brief, plasma samples were collected from white women with normal glucose tolerance (NGT, N = 44) and impaired fasting glucose (IFG, fasting glucose ≥ 5.6 mmol/L, N = 20). The participants were age- and BMI-matched. We limited ourselves to include only individuals with impaired fasting blood glucose as individuals with prediabetes in order to keep the variability of prediabetic pathophysiology small and precise. The study protocols were approved by the relevant local medical ethics committee, and all participants provided informed written consent.

### 4.2. Animal Studies

Five-week-old female New Zealand Obese (NZO/HIBomDife) mice from our own breeding (German Institute of Human Nutrition Potsdam-Rehbruecke, Germany; ethics approval 2347-50-2019) were fed a high fat diet (60%; D12492, Research Diets) until 10 weeks of age. Body weight and blood glucose were monitored weekly and liver fat determined via computer tomography. Diabetes-prone status was assigned based on final blood glucose levels between ≥8.8 mM and <16.6 mM, and liver fat content < 55.2 HU (Hounsfield units). Detailed description of the animals was provided in our previous studies [12,13].

### 4.3. Pancreatic Islet Isolation and RNA-Sequencing

Isolation of islets of Langerhans was performed as described previously [12]. For RNA-sequencing, islets from two to four mice (30-110 islets/mouse) were pooled per sample, resulting in four islet pools from DR mice, and five pools from DP mice. Total RNA was extracted using the miRNeasy Micro Kit (Qiagen, Hilden, Germany) according to manufacturer’s instructions and subsequent DNase treatment. RNA samples with RNA integrity number ≥ 8 (Bioanalyzer, Agilent Technologies, Waldbronn Germany) were used for RNA sequencing. The sequencing was carried out by GATC Biotech (Konstanz, Germany). A detailed protocol and bioinformatic analyses have been previously published [12].

### 4.4. miRNA Detection in Human Plasma

Plasma miRNA of women with and without impaired fasting blood glucose was quantified by using a high-throughput 384-well array and real-time PCR method. Sample preparation, miRNA detection and normalization were carried out by Qiagen (Hilden, Germany). The quality of isolated input RNA was assessed via spike-in assays.

By calculating the ratio between miR-451 as a marker of red blood cells and miR-23a, known to be stably abundant in plasma, hemolysis was ruled out for all the samples used in the study. The data were subject to quality control by melting curve analysis, calculation of amplification efficiency, and comparison of Ct value to background level in the negative control sample. Data normalization was performed based on the average of the assays detected in all samples according to the following formula:$$Normalized\;Ct\;=\;global\;mean\;Ct\left(sample\;1\right)-assay\;Ct\;(miRNA\;of\;interest\;in\;sample\;1)$$

To avoid problems of mixing positive and negative values, a fixed value (+15) was added for all data, and resulting values multiplied by (−1) for more intuitive visualization of the direction of miRNA (up- or downregulated).

### 4.5. miRNA Detection in Mouse Tissues

Total RNA was extracted from mouse pancreatic islets and white adipose tissue (WAT) by using miRNeasy Micro Kit (QIAGEN, Hilden, Germany) according to the manufacturer’s protocol, with additional DNase treatment. Sequencing was carried out by Illumina HiSeq platform, as described previously [12].

### 4.6. miRNA Target Prediction

To define putative target genes of let-7i-5p, we first applied an in silico method based on five different prediction tools, DIANA-microT, miRDB, TarPmiR, TargetScan7.1 and RNA22, whereas optimal targets were given when at least three tools resulted in overlapping target results [12]. The entire list of putative target genes of let7i-5p was then compared to transcripts differentially expressed in pancreatic islets of DP and DR mice. The Kyoto Encyclopedia of Genes and Genomes (KEGG) pathway enrichment analysis was performed using the publicly available Database for Annotation, Visualization and Integrated Discovery (DAVID). The cutoff enrichment score was set to >1.7 and *p*-value < 0.1. The output was filtered for redundant pathways.

### 4.7. Transfection of INS-1 (832/13) β-Cells

To overexpress let-7i-5p, INS-1 (832/13) cells were seeded in 24-well plates and grown to 70% confluency. Cells were transfected with 33 nmol/L of miScript miRNA mimics or the corresponding non-targeting control using Lipofectamine 2000 transfection reagent (Invitrogen, Thermo Fisher Scientific, Waltham, MA, USA) according to manufacturer’s protocol. RNA for expression analyses was collected 72 h post-transfection.

### 4.8. Prediction of Transcription Factor Binding Sites Regulating let-7i-5p

Differentially expressed genes in islets of DP and DR mice were first screened for genes encoding for transcription factors (TFs). Then, only TFs that correlated to let-7i-5p abundance levels in islets were selected, with a threshold of adjusted R^2^ > 0.75. Afterwards, transcription factor binding sites were predicted in an area 2 kb upstream of let-7i using R (version 4.2.3), TFBSTools (version 1.36), and BSgenome.Mmusculus.UCSC.mm10 (version 1.4.3). Finally, binding sites were filtered for full core-motif and fitting genomic strand.

### 4.9. RNA Isolation and Quantitative RT-PCR (qRT-PCR) in INS-1 Cells

Total RNA was extracted from INS-1 cells as described above for mouse tissues. Isolated RNA was subsequently reverse transcribed using M-MLV reverse transcriptase (Promega, Fitchburg, WI, USA) as described [51], and genes of interest were detected via qRT-PCR.

### 4.10. Statistics and Plotting

Statistical significance was analyzed using Welch’s *t*-test. Data are presented as mean ± SEM, with significance indicated by asterisks (* *p* ≤ 0.05, ** *p* ≤ 0.01, *** *p* ≤ 0.001). Where applicable, the number of replicates (N) was stated in the figure legend. Plots were created using R version 3.6, utilizing the RCircos [12,52], DiagrammeR, Graphpad Prism 9, BioRender and gplots packages.

## Figures and Tables

**Figure 1 ijms-24-15673-f001:**
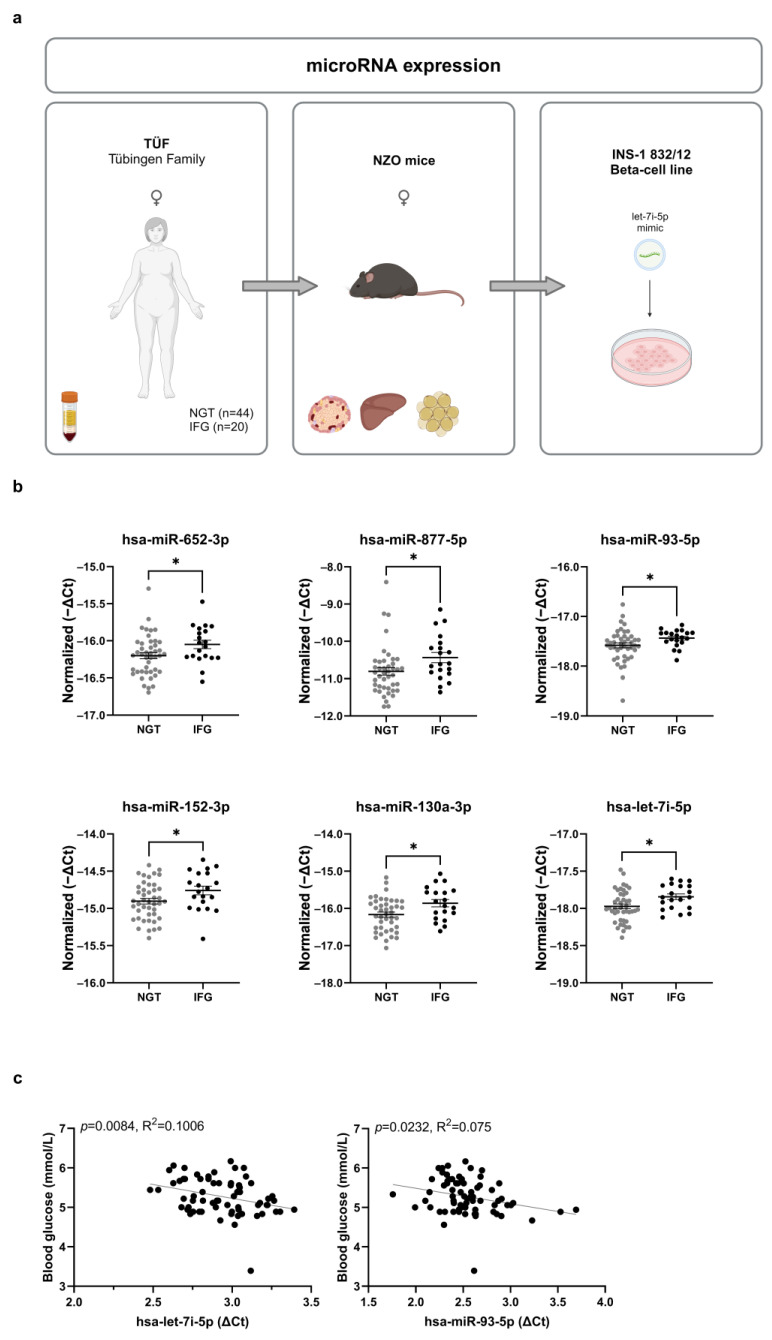
Differences in circulating miRNA levels between women with normal glucose tolerance (NGT) and elevated fasting glucose levels (IFG). (**a**) Study design and summary of experimental approaches. (**b**) Plasma levels of indicated miRNAs significantly different (Welch’s *t*-test) between women with NGT (N = 44) and IFG (N = 20), expressed as −ΔCt. Data are presented as mean ± SEM (* *p* ≤ 0.05). (**c**) Results of the linear regression between let-7i-5p and miR-93-5p levels in plasma, and fasting blood glucose (N = 64).

**Figure 2 ijms-24-15673-f002:**
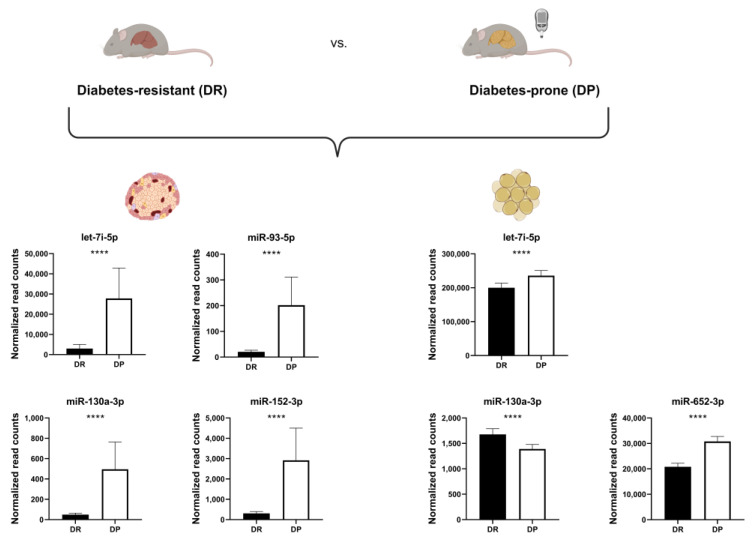
MiRNA levels in islets and adipose tissue of diabetes-prone (DP) NZO mice are similar to those of women with elevated fasting glucose. Levels of putative prediabetes miRNA markers in islets (left) and WAT (right) of diabetes-prone (N = 15) and diabetes-resistant (DR; N = 15) mice, expressed as normalized read counts. Data are presented as mean ± SEM (Welch’s *t*-test; **** *p* ≤ 0.0001).

**Figure 3 ijms-24-15673-f003:**
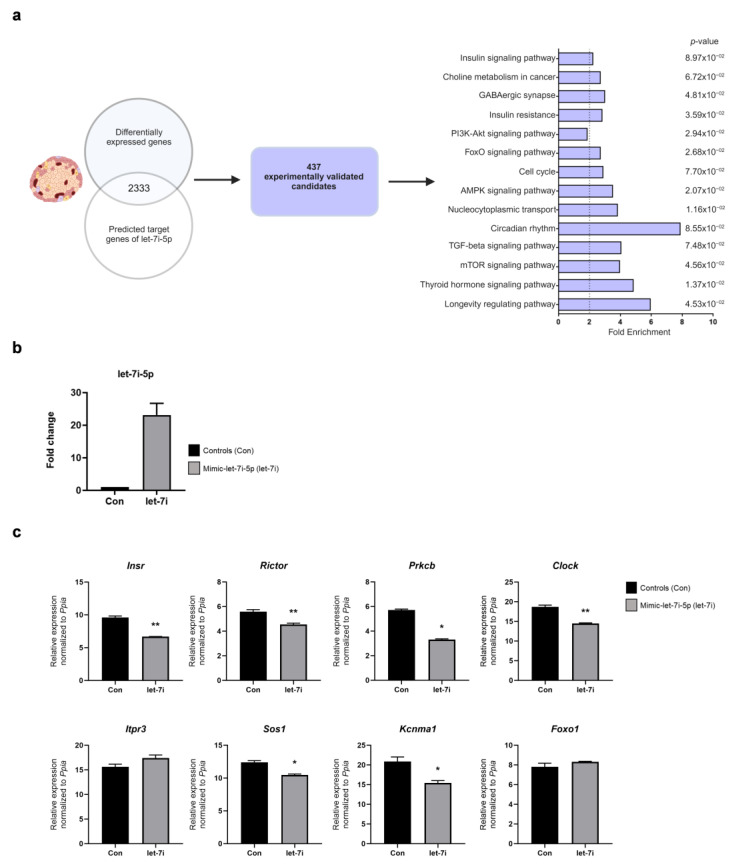
Let-7i-5p overexpression in INS-1 832/12 cells reduces expression of genes involved in the insulin signaling pathway. (**a**) Comparison of the differentially expressed transcripts with predicted and experimentally validated genes (left panel). The bar plot shows the enriched pathways among genes downregulated in pancreatic islets of DP mice, compared to DR mice, which are additionally found in databases (e.g., DIANA) as experimentally validated targets of let-7i-5p in humans (right panel). (**b**) Fold change in let-7i-5p expression in INS-1 cells 72 h after transfection with mimic-let-7i-5p. (**c**) mRNA expression levels of indicated genes, candidates from insulin signaling pathways, after let-7i-5p overexpression in INS-1 cells. Data were obtained from at least three wells per condition from three independent experiments. Data are presented as mean ± SEM (Welch’s *t*-test; * *p* ≤ 0.05, ** *p* ≤ 0.01).

**Table 1 ijms-24-15673-t001:** Participants’ characteristics in the NGT and IFG groups. Data are shown as mean values with standard deviation. Asterisks indicate a significant difference between NGT and IFG participants with Welch’s *t*-test (**** *p* < 0.0001).

	NGT	IFG
N	44	20
Age (years)	49.8 ± 11	49 ± 13.0
BMI (kg/m^2^)	32 ± 5	31 ± 3.6
Fasting blood glucose (mmol/L)	5.0 ± 0.2	5.8 ± 0.2 ****
Blood glucose 120 min after glucose bolus during OGTT (mmol/L)	6.0 ± 0.9	6.3 ± 0.8

## Data Availability

All relevant data generated or analyzed during this study are included in this manuscript. Raw data were deposited under gene accession number GSE143875, GSE208630. Further datasets used and/or analyzed during the current study are available from the corresponding author upon reasonable request.

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
