# Peer review of "Identification of MicroRNAs Associated with Prediabetic Status in Obese Women"

_ijms, 2023, doi:10.3390/ijms242115673_

Round 1

Reviewer 1 Report

Comments and Suggestions for Authors

This is an interesting study that identifies potential circulating miRNA markers for pre-diabetes. The authors additionally show that these miRNA have functional effects on insulin sensitivity and pancreatic function in translational studies in animals and cells in vitro.

My concern, though, is that the authors don't definitively show that the miRNAs have diagnostic potential. Particularly in relation to normalization of the human data and the specificity of the markers. The authors should comment on the reliability of their finding.

Comments

Was isolated RNA quality controlled?

Data expressed as deltaCt suggests normalization was performed, but it is unclear what reference was used. It is important that the reference is unaffected by the IFG. Please elaborate.

Data plotted as deltaCt, but this is an exponential parameter. What was the efficiency of the (RT)PCR? This could be taken into account, and may even improve the effect size.

Data could also be expressed as delta-delta Ct (Ctrls v IFG).

Reviewer 2 Report

Comments and Suggestions for Authors

The authors presented an interesting evaluation of circulating and tissue miRNA expression in patients or models of prediabetes.

They identified a set of miRNAs increasing in plasma of women with impaired fasting glucose levels (IFG) when compared to controls and the mouse homolog of some of these miRNAs was observed as dysregulated in pancreatic and adipose tissue of a mouse model of the disease.

The work is interesting and well-designed; however, some aspects should be addressed:

-       Please convert the delta CT of Figure 1b to gain a coherent representation of the results. Despite miRNAs being upregulated in IFG using the deltaCT, the plot seems to represent a downregulation. Similarly, the results of panel c must be properly reported. Using the minus deltaCT should be enough.

-       Despite no difference in BMI and age observed among subject groups as reported in Table 1, the Authors must test a possible significant relationship between miRNA levels and these covariates. A multivariate analysis using a generalised linear model using miRNA levels as the dependent variable and age, BMI, and subject class is enough to investigate this candidate dependency.

-       Based on at least public data analysis, the authors should test their findings in a validation cohort. Several microarray or sequencing experiments were performed on the plasma of prediabetic subjects, for example, GSE148961, GSE186883, and GSE147965. The authors must verify whether the identified miRNAs are differentially expressed, or at least upregulated in these datasets.

-       Is completely unclear how the RNA-Seq data and differentially expressed transcript reported in chapter 2.3 were obtained and no details in the Materials and Method section are reported. Please provide all the details needed to allow the reader to completely reproduce the reported results.

-       Is not clear the order criteria used for reporting the enrichment analysis results in Figure 4 panel a. Without any bias it would be better to just order the results by decreasing fold enrichment.

-       It would be interesting to investigate the list of differentially expressed genes obtained in the analysis. For example, the authors should perform an enrichment analysis on these genes and evaluate whether any transcriptional factors regulating the expression of candidate miRNAs (particularly let-7i-5p) are among the differentially expressed genes.

-       The pathway enrichment analysis reported in Figure 4 a should be reported by dividing between upregulated and downregulated genes. Is the target of let-7i-5p prevalent in the downregulated genes or not?

-       It would be very interesting to investigate or at least discuss the role of the estrogen receptor signalling pathway in the regulation of these miRNAs.

Round 2

Reviewer 1 Report

Comments and Suggestions for Authors

The authors have adequately addressed my comments.

Reviewer 2 Report

Comments and Suggestions for Authors

The authors addressed all the points raised during the revision process.